# Silence as a Quiet Strategy: Understanding the Consequences of Workplace Ostracism Through the Lens of Sociometer Theory

**DOI:** 10.3390/bs15081022

**Published:** 2025-07-28

**Authors:** Jun Yang, Bin Wang, Yijing Liao, Feifan Yang, Jing Qian

**Affiliations:** 1School of Economics and Management, Tongji University, 1 Zhangwu Street, Yangpu, Shanghai 200092, China; 2Department of Management, Monash University, 7 Sir John Monash Drive, Caulfield East, Melbourne, VIC 3145, Australia; atomwong@126.com; 3Future of Work Institute, Curtin University, 78 Murray Street, Perth, WA 6000, Australia; yijing.liao@curtin.edu.au; 4Business School, Soochow University, 50 East Ring Road, Suzhou 215006, China; ffyang@suda.edu.cn; 5Business School, Beijing Normal University, 19 Xinjiekouwai Street, Beijing 100875, China; jingqian@bnu.edu.cn

**Keywords:** workplace ostracism, defensive silence, organization-based self-esteem, sense of power, sociometer theory

## Abstract

Existing research has predominantly framed defensive silence as an avoidance response to interpersonal mistreatments. Moving beyond this view, this study theorizes defensive silence as a proactive strategy for managing interpersonal relationships through the lens of sociometer theory. We posit that workplace ostracism will reduce employees’ organization-based self-esteem (OBSE), which in turn increases their subsequent defensive silence to avert further damage to relationships. In addition, we also expect a moderating role of the sense of power in mitigating the negative impact of workplace ostracism on OBSE. Based on the multi-wave, multi-source data of 345 employees and their 82 immediate supervisors, we tested all the hypotheses. Results from multilevel modeling indicated that OBSE mediated the indirect effect of workplace ostracism on defensive silence, and also supported the moderation role of sense of power. Our theoretical model provides a novel perspective that deepens the understanding of defensive silence and suggests implications for managerial practices.

## 1. Introduction

Defensive silence refers to the intentional withholding of potentially important information, suggestions, and ideas ([9]; [25]). Such silence obstructs open dialog, which is vital for effective problem solving and overall organizational performance. In Chinese workplaces, where cultural norms such as maintaining interpersonal harmony and respecting hierarchical authority discourage open dissent, the consequences of defensive silence are even more severe ([30]).

Prior research has primarily viewed defensive silence as an emotionally driven avoidance response to workplace mistreatment, such as ostracism ([16]; [17]; [18]). These scholars have suggested that defensive silence is primarily driven by emotional mechanisms, particularly the fear of being excluded by coworkers, leading individuals to withhold their voices as a form of self-protection ([3]). However, this emotional framing may overlook the deliberate and strategic nature of silence in some contexts. For instance, employees may also choose to remain silent as a proactive means of navigating workplace dynamics after carefully evaluating the interpersonal risks and benefits of speaking up ([9]; [14]). Moving beyond the predominant emotional mechanisms (e.g., fear), the current study aims to understand the relationship between workplace ostracism and defensive silence through the lens of sociometer theory.

Sociometer theory explains how individuals monitor and respond to social inclusion ([35]; [41]). According to this theory, the self-esteem system is not merely a self-evaluation tool, but functions as an internal gauge that continuously tracks the extent to which one feels accepted and valued by others. Meanwhile, this system plays an adaptive function in regulating behavior to maintain adequate levels of interpersonal acceptance ([6]; [15]). A core assumption of sociometer theory is that individuals are motivated to preserve a sense of belonging and significance within social and work relationships, as doing so is essential for accessing the resources needed to function effectively at work ([19]). Self-esteem, therefore, fluctuates in response to perceived social inclusion or exclusion, helping individuals adjust their behavior to maintain group membership and relational security ([24]). In organizational contexts, being ostracized by coworkers not only conveys the message that one is not seen as a valuable organizational member but threatens the conditions necessary for effective work ([15]; [38]). In response, the sociometer system of self-esteem perceives a reduction in their importance, value, and effectiveness as a member of the organization (i.e., organization-based self-esteem, OBSE) ([15]).

As a result, the decline in OBSE functions as an alarm signal, regulating behavior to maintain essential levels of social acceptance ([6]; [23]). In particular, a diminished OBSE prompts individuals to seek acceptance from their colleagues. This, in turn, strongly motivates them to refrain from any behaviors or comments that might exacerbate their experience of ostracism ([36]). Hence, individuals may opt for silence, withholding ideas, viewpoints, and information to prevent further deterioration of work relationships. Accordingly, we propose that workplace ostracism adversely affects employees’ OBSE, leading them to engage in defensive silence to avert further relational damage.

In addition, sociometer theory emphasizes that the extent to which ostracism damages self-esteem depends on the instrumental value of the ostracizer in achieving important goals ([29]; [35]). This highlights the role of dependence in organizational relationships, shaping how individuals assess and respond to relational threats such as exclusion. Building on this premise, we examine the moderating role of the sense of power. Sense of power refers to an individual’s subjective perception of their capacity to influence others and access resources within an organizational context ([39]; [49]). It reflects an individual’s perceived autonomy and dependence in social interactions, which fundamentally shapes their sensitivity to exclusion and their capacity to protect self-worth under threat ([39]). Specifically, individuals with a strong sense of power see themselves as less dependent on others for achieving goals or securing resources. This perceived autonomy enables them to maintain a sense of control and resilience in the face of ostracism ([22]). Consequently, they feel less dependent on social affirmation for access to resources and goal attainment, placing less emphasis on coworker acceptance, which mitigates the adverse effects of ostracism on their OBSE. In contrast, individuals with lower perceived power depend heavily on others for resources and support, heightening their sensitivity to relational threats such as ostracism. Ostracism, for these individuals, represents a significant loss of access to critical resources, thereby exacerbating its detrimental impact on OBSE ([39]). Accordingly, the sense of power moderates the relationship between workplace ostracism and OBSE, with high-power individuals demonstrating greater resilience and low-power individuals exhibiting heightened vulnerability.

As depicted in Figure 1, this study, drawing on the sociometer theory, proposes that workplace ostracism will positively lead to defensive silence via decreased OBSE. In addition, we also expect that the effect of workplace ostracism on OBSE will be stronger among employees with a weak sense of power. In the following sections, we will explicate our research model, conduct formal tests of our hypotheses, and discuss key findings and implications for managerial practice.

## 2. Theoretical Background and Hypotheses

### 2.1. The Mediating Role of OBSE

OBSE is defined as employees’ self-perception about their importance, worthiness, meaningfulness, and effectiveness within their organization ([11]; [37]). According to previous research, identity acquisition and interpersonal acceptance significantly predict employees’ OBSE ([36]). For example, the approval of the role identity, whether from supervisors or colleagues, provides a form of positive feedback that they are performing well in the organization, which fosters their sense of meaningfulness and achievement ([5]). In addition, interpersonal acceptance or rejection is a direct signal of how others regard and interact with them, conveying to employees their value and importance to the organization, which is also a primary element in shaping their OBSE ([15]; [34]). The sociometer theory highlights that the quality of interpersonal relations at work is a crucial situational factor that affects workers’ self-esteem because it is associated with perceived social acceptance and identity acquisitions ([15]). Workplace ostracism at work includes silent treatment, giving the cold shoulder, avoiding conversation or eye contact, and withholding needed information ([41]; [46]), indicating that a person has fallen into a detrimental workplace relationship in the workplace. Consistent with this, we posit that workplace ostracism has a negative effect on OBSE.

According to sociometer theory, such experiences activate an internal system that monitors social acceptance and self-worth. When individuals are socially excluded, they are less likely to receive recognition, feedback, or validation related to their role identity. This lack of social input undermines their sense of meaning and accomplishment at work, ultimately lowering their organization-based self-esteem (OBSE).

Specifically, ostracism is inherently noninteractive, involving behaviors such as avoiding interaction or excluding individuals from conversations ([50]). In addition, considerable studies have demonstrated that workplace ostracism is a negative interpersonal signal, implying that one is unwelcomed, unaccepted, and part of the out-group ([5]; [8]). These signals convey to employees that they are not valuable members of the organization, diminishing their self-perception of importance and worthiness ([48]). As a result, their self-esteem at work decreases ([38]; [50]). In addition, the deterioration of social relations severely conditions individuals’ work effectiveness, because being well-connected with colleagues or customers not only offers them access to support and crucial work-related information but also provides them with models to learn by observing how others tackle problems. Conversely, being excluded from social networks prevents them from exchanging information, accessing resources, and seizing learning opportunities. This limitation substantially diminishes their work effectiveness and perceived potential for success, which are also vital contributors to OBSE ([50]).

In sum, employees who experience ostracism are likely to suffer damage to their perceived worthiness, meaningfulness, and work effectiveness, which are essential components of organization-based self-esteem (OBSE). Therefore, we propose the following hypothesis:

**Hypothesis** **1:**
*Workplace ostracism is negatively related to OBSE.*


Defensive silence is an intentional and strategic behavior aimed at protecting oneself from potential threats ([4]; [31]). Previous research has shown that employees often adopt this type of silence for two main reasons. First, it serves to minimize hostile interactions and prevents being labeled negatively, helping employees restore relational ties and reinforcing their importance within the organization ([42]); second, this behavior enables employees to reduce interpersonal barriers to effective work, thereby regaining a sense of effectiveness and control.

Drawing on sociometer theory, the drop in the esteem system instinctively activates a coping mechanism to reduce the likelihood of triggers that could intensify social exclusion ([24]). Specifically, similarly to reactions toward physical danger, lower levels of OBSE signal a heightened risk of impending or ongoing exclusion from a group ([10]). As an alarm, it prompts employees to seek acceptance to restore their perceived importance and to receive more work-related information and support ([28]). In order to minimize the risk of future rejection and to be integrated into desirable relationships, they become keenly aware of the potential negative repercussions of speaking up, and avoid any actions or remarks that might be seen as problematic. As a result, they tend to consciously withhold their ideas, viewpoints, and facts ([18]; [44]).

In contrast, individuals with higher levels of OBSE perceive they are valued and competent within their workplace. In other words, their sociometers detect no threats to their inclusion, which fundamentally shapes their behavioral responses. Much like a car with a full tank of gas, individuals with higher levels of OBSE operate without constant concern over their interpersonal standing or a need to monitor their “fuel gauge” closely. Boosted by the full interpersonal tank, they feel no urgency to go out of their way to purposely seek endorsements and recognition ([23]). As a result, those individuals high on OBSE focus on expressing their thoughts and opinions openly without apprehension or the need to withhold information, leading to a lower incidence of defensive silence. We, therefore, hypothesize as follows:

**Hypothesis** **2:**
*OBSE is negatively related to defensive silence.*


Based on sociometer theory, this study regarded OBSE as a sociometer for monitoring interpersonal relationships, which, in turn, influences employees’ behaviors at work. Specifically, we argue that ostracized employees tend to exhibit lower levels of OBSE, and this reduction in OBSE serves as a warning, prompting them to adopt defensive silence to avert further damage to relationships. Accordingly, we propose the mediating role of OBSE in the relationship between workplace ostracism and defensive silence:

**Hypothesis** **3:**
*OBSE mediates the effect of workplace ostracism on defensive silence.*


### 2.2. The Moderation of Sense of Power

According to the sociometer theory and the well-established ostracism literature, the psychological impact of social exclusion is shaped by the perceived importance of the relationship with the ostracizer ([7]; [24]). In workplace contexts, colleagues who hold instrumental value, such as providing support, collaboration opportunities, or access to information, can strongly influence how individuals experience and respond to exclusion. Sense of power refers to an individual’s subjective perception of their capacity to influence others and control valued outcomes in social and organizational interactions ([20]; [49]). A strong sense of power enhances individuals’ perceived control, increases their self-importance, and reduces the psychological weight they place on being accepted or supported by others. Accordingly, we propose that the sense of power diminishes the negative impact of workplace ostracism on OBSE.

Specifically, individuals with a high sense of power are more likely to view themselves as self-sufficient and less reliant on interpersonal relationships to access resources, gain recognition, or maintain work effectiveness ([20]; [26]). As a result, they may interpret exclusionary behaviors from coworkers as less threatening or personally significant ([21]). Even when facing workplace ostracism, these individuals tend to maintain confidence in their capabilities and worth ([12]), leading to a relatively low reduction in OBSE.

In contrast, individuals with a weak sense of power tend to perceive themselves as dependent on others to meet work-related needs, such as receiving information, social support, or development opportunities, which leads them to focus more on how they are treated within groups ([29]; [39]). For these employees, being excluded may signal not only social rejection but also a potential loss of tangible resources, career support, and team belonging, all of which are critical for maintaining a positive organizational identity. As a result, they are more vulnerable to declines in OBSE, as exclusion directly undermines their perceived social worth and professional standing ([24]; [23]). Thus, we propose the following:

**Hypothesis** **4:**
*Sense of power moderates the negative effect of workplace ostracism on employees’ OBSE such that the relationship is weaker when employees’ sense of power is strong.*


## 3. Materials and Methods

### 3.1. Participants and Procedures

Data were collected from a large state-owned travel company in Shanghai. With the help of the human resources department, we introduced the purpose and procedures of our research to all staff members attending a regular meeting of each department, inviting their voluntary participation in this project. Considering participants’ concerns for privacy and confidentiality, we promised that all data collected would be anonymized and would only be used for research purposes.

To mitigate the risk of common method variance, we collected data in three separate waves. This multi-wave approach was intended to create temporal distance between measurements, thus reducing potential biases associated with single-time-point reports. In addition, we asked team leaders, rather than employees themselves, to rate defensive silence to minimize self-report bias. Specifically, in the first survey (Time point 1), employees were asked to report their demographic information, perceived workplace ostracism, and sense of power. A month after the first survey (Time point 2), participants were asked to report their OBSE. Two months after the first survey (Time point 3), team leaders were asked to evaluate the defensive silence of their subordinates. Each participant was given a unique identification number to facilitate the matching of questionnaires collected by different teams across three waves of surveys. To encourage employees to participate in and complete this project, we offered a book (worth 50 RMB) to those who completed all three waves of the survey. Throughout the survey, two research assistants were employed to help distribute and collect the questionnaires.

During the first wave of data collection, we distributed 590 questionnaires to the employees attending department meetings, receiving 523 valid employee questionnaires from 105 teams for a response rate of 88.64%. One month later, we distributed questionnaires to the 532 employees who completed the first wave and received 412 responses from 95 teams for a response rate of 78.78%. In wave 3, we collected the supervisor’s assessment of defensive silence for each responding employee in their work team. The final valid sample consisted of 345 employees and 82 supervisors, with a response rate of 83.74%. According to G*Power analysis (v3.1.9.7; f^2^ = 0.15, power = 0.95), 138 participants were required; our sample of 345 thus ensures sufficient power for hypothesis testing. Of the 345 employees, 61.45% were female, and 38.55% were male. The average age was 33.06 years (SD = 7.70). Further, 77.1% received a college degree or junior college degree, and the average organizational tenure was 3.74 years (SD = 3.79).

### 3.2. Measures

The survey items were translated into Chinese using a translation-back translation procedure to ensure equivalence. A pilot test confirmed the clarity and cultural relevance of the items. Except for the measurement of workplace ostracism, all ratings were based on a seven-point Likert-type scale ranging from 1 (strongly disagree) to 7 (strongly agree).

***Workplace ostracism.*** Workplace ostracism was measured by a ten-item scale developed by [10] ([10]). The sample item is “My greetings have gone unanswered at work.” Items were rated on a seven-point Likert scale ranging from 1 (Never) to 7 (Always). Cronbach’s alpha of this scale was 0.97.

***Organization-based self-esteem (OBSE)*.** We measured OBSE with a ten-item scale developed by [37] ([37]). The sample item is “I can make a difference in my team.” Cronbach’s alpha of this scale was 0.83.

***Sense of power*.** Sense of power was measured by an eight-item scale developed by [1] ([1]). The sample item is “I can get others to do what I want.” Cronbach’s alpha of this scale was 0.82.

***Defensive silence*.** Defensive silence was measured by a five-item scale developed by [9] ([9]). Supervisors were asked to evaluate the extent to which each statement described the work status of their subordinates. The sample item is “This employee doesn’t express suggestions for change to self-protection.” Cronbach’s alpha of this scale was 0.96.

***Control variables*.** Following the well-established ostracism literature, age and gender may impact individuals’ sensitivity to social ostracism; meanwhile, individuals with higher education levels may exhibit greater esteem when appraising ostracism experiences or possess better coping strategies ([16]). Moreover, employees with longer tenure generally have deeper social integration within their teams, potentially buffering them from negative ostracism consequences ([33]). Therefore, we controlled for participants’ age, gender (coded 1 for “male” and 2 for “female”), education, and team tenure. We also tested the models after removing the control variables, and the results showed no significant differences.

### 3.3. Analytic Strategy

Given the nested nature of our data (i.e., individuals were nested in different teams), we constructed a null model via hierarchical linear modeling (HLM) to calculate the variance between groups. This approach was chosen because it accounts for the interdependence of observations within teams, thereby accurately estimating both within- and between-group effects. Our data include 82 teams, which exceeds the commonly recommended minimum of about 50 clusters for accurate estimation in multilevel analyses ([13]; [27]; [32]). We constructed a null model via HLM to calculate the variance between groups. The results showed the between-group variance percentages for workplace ostracism, OBSE, sense of power, and defensive silence were 28.4%, 34.0%, 37.3%, and 64.0%, respectively, indicating significant team-level variability. Therefore, we used Mplus 7.4 to test our hypotheses with a nested structure. Further, all predictors were group-mean centered.

We constructed three models for hypothesis testing. Model 1 (i.e., random coefficient model) examined Hypotheses 1 and 2 by regressing OBSE on workplace ostracism and defensive silence on both. For Hypothesis 3, the 1-1-1 mediation was tested by multiplying the random slopes of ostracism → OBSE and OBSE → silence. Model 2 tested Hypothesis 4 by regressing OBSE on the interaction between ostracism and sense of power. Model 3 assessed the moderated mediation effect.

## 4. Results

### 4.1. Descriptive Statistics

Table 1 presents the means, standard deviations, and correlations for the variables in our study. As shown in Table 1, workplace ostracism was negatively correlated with OBSE (r = −0.377, *p* < 0.001). In addition, OBSE was also negatively correlated with defensive silence (r = −0.123, *p* < 0.05). The findings supported our next step of mediating effects analysis of OBSE.

### 4.2. Confirmatory Factor Analysis

We conducted a confirmatory factor analysis (CFA) to assess the discriminant validity of the four variables. Specifically, we compared the hypothesized four-factor model against a series of alternative models. The results indicated that the four-factor model provided a better fit to the data (χ^2^/df = 2.77, RMSEA = 0.05, CFI = 0.90, NFI = 0.86, IFI = 0.91, TLI = 0.88) than the three-factor model (χ^2^/df = 5.92, RMSEA = 0.09, CFI = 0.71, NFI = 0.69, IFI = 0.71, TLI = 0.67), the two-factor model (χ^2^/df = 9.72, RMSEA = 0.15, CFI = 0.63, NFI = 0.62, IFI = 0.64, TLI = 0.59), and the one-factor model (χ^2^/df = 12.69, RMSEA = 0.19, CFI = 0.46, NFI = 0.45, IFI = 0.37, TLI = 0.39). These results indicated that the fit of the hypothesized model was relatively better than the alternatives, supporting the discriminant validity of the constructs.

### 4.3. Hypothesis Testing

Hypotheses 1 and 2 were tested in Model 1. As shown in Table 2, workplace ostracism was negatively related to OBSE (B = −0.500, SE = 0.089, *p* = 0.000), and OBSE was negatively associated with defensive silence (B = −0.772, SE = 0.059, *p* < 0.000). Therefore, Hypotheses 1 and 2 were supported.

Hypothesis 3 proposed a mediation effect, whereby OBSE plays a mediating role in the indirect effect of workplace ostracism on defensive silence. Our results showed that the indirect effect was significant (indirect effect = 0.386, SE = 0.060, *p* = 0.03, 95% CI = [0.028, 0.137]), providing support for Hypothesis 3.

Hypothesis 4 predicted that sense of power moderates the effect of workplace ostracism on OBSE. As shown in Table 3, the interaction term (ostracism × sense of power) was significant (B = 0.097, SE = 0.046, *p* = 0.035). Simple slope analyses (Figure 2) revealed that the negative effect of ostracism on OBSE was stronger for employees with a low (−1 SD; B = −0.599, SE = 0.106, *p* = 0.000) rather than high (+1 SD; B = −0.404, SE = 0.095, *p* = 0.000) sense of power. The difference between slopes was also significant (ΔB = −0.153, SE = 0.071, *p* < 0.033), supporting Hypothesis 4.

To further test moderated mediation, we examined whether the indirect effect of ostracism on defensive silence via OBSE varied by sense of power. The result shows that the indirect effect was stronger for low-power employees (indirect effect = 0.470, SE = 0.078, *p* = 0.000, 95% CI = [0.342, 0.598]). The difference in mediation between high and low power was also significant (Δ = −0.154, SE = 0.074, *p* = 0.038, 95% CI = [−0.270, −0.035]), confirming the moderated mediation effect.

### 4.4. Supplementary Analysis

Defensive silence was verified in this study as a coping strategy for workplace ostracism, and we further extended its investigation to ascertain whether defensive silence effectively reduces ostracism. Therefore, we further measured workplace ostracism at Time Point 3 using the same scale and estimated an additional model. This model regressed OBSE *_t_*_2_ on workplace ostracism *_t_*_2_, regressed defensive silence *_t_*_3_ on OBSE *_t_*_2_, and regressed workplace ostracism *_t_*_3_ on defensive silence *_t_*_3_. Results from our multilevel analysis indicated that while workplace ostracism *_t_*_2_ was negatively related to OBSE *_t_*_2_ (B = −0.408, SE = 0.071, *p* = 0.000) and OBSE *_t_*_2_ was negatively associated with defensive silence *_t_*_3_ (B = −0.730, SE = 0.045, *p* = 0.000); the coefficient of the “defensive silence *_t_*_3_ → workplace ostracism *_t_*_3_” regression was not significant (B = −0.138, SE = 0.104, ns). This suggests that defensive silence does not effectively reduce workplace ostracism. One possible reason is that while it is meant as a protective strategy, defensive silence may not be perceived by coworkers as a signal of resolution or reconciliation. Instead, it may maintain social distance and be perceived by coworkers as disengagement, lack of interest, or even passive resistance, thereby failing to change the ostracizers’ perceptions or behaviors.

## 5. Discussion

To examine the consequences of workplace ostracism through the lens of sociometer theory as well as the moderating role of the sense of power, we conducted a multi-wave, multi-source field study. Based on data from 345 employees and their 82 immediate supervisors, our multilevel modeling results revealed that workplace ostracism has a positive indirect effect on defensive silence through employees’ OBSE. In addition, we found that sense of power mitigated the impact of workplace ostracism on OBSE, such that the effect of workplace ostracism on OBSE was weaker for employees with a stronger sense of power.

### 5.1. Theoretical Implications

The current study has several important theoretical implications. First, it expands the literature on defensive silence by framing it not only as a fear-based reaction ([18]), but also as a proactive, self-regulatory strategy involving rational judgment ([47]). This perspective highlights defensive silence as a deliberate choice in navigating complex workplace dynamics. Additionally, although defensive silence was conceptualized as a coping response to ostracism, our supplementary analysis found it did not reduce exclusion. This finding reveals that while defensive silence may help individuals avoid conflict or emotional exposure, it may also inhibit social dialog and resolution, thereby limiting its capacity to restore social inclusion. In addition, prior research suggests that ostracism is shaped by personal traits, leadership, and context ([16]; [29]). A more comprehensive model is needed to capture these influences.

Second, our study contributes to sociometer theory by clarifying how workplace exclusion affects employees’ self-esteem and subsequent behaviors. Using sociometer theory, we shift the emphasis from emotional distress to understanding self-esteem as an indicator of social acceptance. Our findings show that when employees feel excluded or marginalized at work, their organizational self-esteem decreases, prompting them to adopt silence as a protective response. Thus, we confirm that self-esteem acts as a social alarm: experiencing exclusion signals diminished social value, triggering behaviors aimed at self-protection. Our research broadens sociometer theory by applying it specifically to workplace contexts involving exclusion and employee silence.

Moreover, we explored the moderator role of sense of power in the relationship between workplace ostracism and employees’ OBSE, underscoring individual differences in sensitivity to social cues. While sociometer theory suggests variability in sensitivity to social signals ([35]), direct empirical evidence remains limited. Our study addresses this gap by demonstrating that an employee’s sense of power serves as a key boundary condition: individuals who perceive themselves as having greater power exhibit a stronger relationship between workplace exclusion and reduced self-esteem. This result integrates sociometer theory with research on power dynamics, indicating that psychological resources such as power enhance organizational self-esteem. Thus, our work not only offers a more nuanced understanding of how self-esteem operates in social contexts, but also broadens sociometer theory by illustrating that the effectiveness of self-esteem as a social indicator depends on employees’ perceived influence within organizational hierarchies.

### 5.2. Practical Implications

This study offers several practical implications. Our findings indicate that workplace ostracism undermines employees’ OBSE and enhances defensive silence, underscoring the need for organizations to foster open and supportive climates. To address this issue, organizations should implement clear practices that actively promote inclusion and appreciation. For instance, managers could regularly conduct anonymous surveys or facilitate structured team feedback sessions to assess employees’ sense of belonging. These measures enable the early detection of exclusionary dynamics and timely supportive interventions ([40]). Organizations can further enhance employees’ self-esteem by involving them in decision-making processes and implementing recognition programs such as peer-nominated awards, clearly signaling their value within the team ([2]). Additionally, social skills training and team-building activities can strengthen interpersonal relationships and reduce the occurrence of ostracism. Managers should also empower employees by providing greater autonomy and authority, helping to buffer the negative effects associated with perceived exclusion ([43]). Finally, organizations can support marginalized employees by establishing assistance programs or ombuds offices. Regularly assessing employees’ sense of belonging and including inclusion criteria in managerial performance reviews further signals that employee contributions and well-being are valued.

Our findings also offer valuable guidance for employees. Individuals can take steps to affirm their own inclusion. We suggest that employees actively engage in communication and relationship building. For instance, speaking up in meetings or joining employee resource groups can provide positive social feedback. In addition, the supplementary analysis showed that defensive silence does not significantly reduce workplace ostracism. While it may serve as a coping mechanism, it can increase marginalization and prove harmful over time. Therefore, employees facing ostracism should adopt more proactive strategies to manage workplace relationships. Enhancing social skills—such as communication, interpersonal interaction, and problem-solving—can improve engagement ([45]). Rather than withdrawing, employees are encouraged to actively participate in social activities to strengthen workplace connections.

## 6. Study Limitations and Future Directions

This study has several limitations. First, data were collected from a single Chinese company, which, while controlling for contextual factors, limits the generalizability of findings to other cultures and industries. Cultural differences may also influence how individuals perceive their value and power. For example, due to a stronger focus on interpersonal harmony, Chinese employees may be more inclined to adopt silence in response to ostracism, and the effects of power may be less pronounced in cultures that de-emphasize self-assertion ([7]). Future studies should validate these findings across diverse cultural and organizational contexts.

Second, although we adopted a sociometric perspective to examine defensive silence, we did not control for emotional mechanisms such as fear, nor did we include stable psychological traits like attachment styles or Big Five personality factors. These factors may confound the effects of OBSE and silence, and future research should incorporate them as control or moderating variables to improve model clarity and robustness.

Third, this study employed a time-lagged design. While this approach helps reduce common method bias, this design limits the ability to capture dynamic changes and causal pathways. Future studies should consider adopting fully longitudinal designs (such as the experience sampling method) to better explore how this relationship unfolds over time.

In addition, while our analyses reveal statistically significant relationships among key variables, the relatively small effect sizes observed suggest that other unmeasured variables may be influencing the outcomes. To address this, future research should aim to incorporate a broader range of covariates, including but not limited to individual personality traits, perceptions of organizational climate, leadership support mechanisms, and organizational scale. These additions could help mitigate omitted variable bias and yield more robust conclusions.

## Figures and Tables

**Figure 1 behavsci-15-01022-f001:**
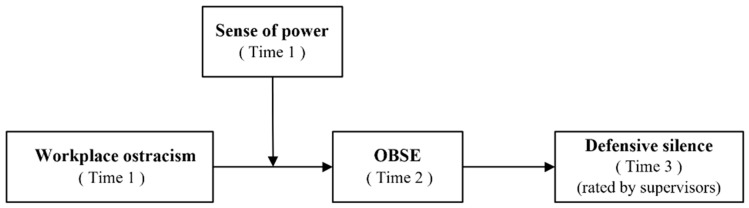
Theoretical model.

**Figure 2 behavsci-15-01022-f002:**
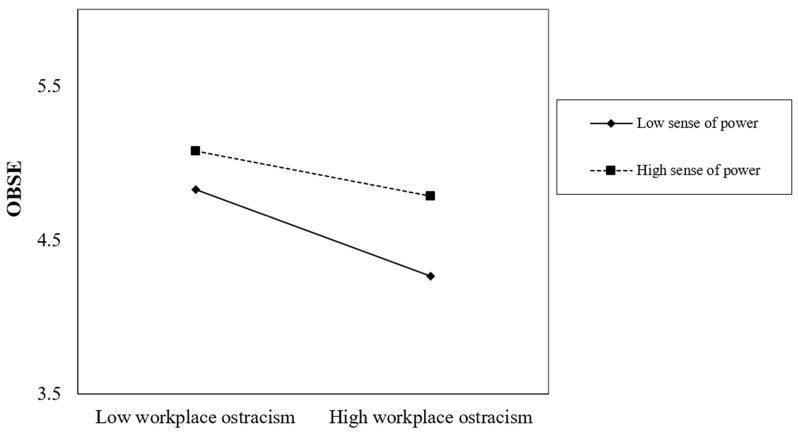
The moderating effect of the sense of power.

**Table 1 behavsci-15-01022-t001:** Descriptive statistics and correlations among study variables.

	Mean	SD	1	2	3	4	5	6	7	8
1. Gender	1.61	0.48								
2. Age	33.06	7.70	−0.077							
3. Edu	3.52	0.93	−0.092	−0.132 *						
4. Team tenure	3.92	4.13	0.018	0.346 **	−0.139 **					
5. Workplace ostracism	1.82	0.89	−0.014	0.054	−0.030	0.067	(0.97)			
6. OBSE	5.34	1.00	−0.073	−0.101	0.175 **	−0.085	−0.377 **	(0.83)		
7. Defensive silence	2.42	1.26	−0.005	0.034	0.101	0.000	0.158 **	−0.123 *	(0.96)	
8. Sense of power	3.67	1.03	−0.108 *	0.054	−0.110 *	0.063	0.239 **	−0.067	−0.005	(0.82)

Note: Coefficient alpha estimates of the reliability are in parentheses on the diagonal. * *p* < 0.05, ** *p* < 0.01.

**Table 2 behavsci-15-01022-t002:** Results of multilevel path analysis for testing Hypotheses 1–2 (model 1).

	OBSE			Defensive Silence		
	B	SE	β	B	SE	β
Gender	0.006	0.070	0.003	−0.013	0.044	−0.005
Age	−0.001	0.006	−0.007	0.006	0.005	0.037
Education	0.008	0.030	0.007	−0.068	0.040	−0.050
Team tenure	−0.031	0.009	−0.128	0.014	0.009	0.046
Workplace ostracism	−0.500 **	0.089	−0.445	0.030	0.053	0.021
OBSE				−0.772 **	0.074	−0.613 **
R^2^	0.651			0.903		

Note: N_level-1_ = 345, N_level-2_ = 82; Bs represents unstandardized regression coefficients from path-analytical modeling. ** *p* < 0.01.

**Table 3 behavsci-15-01022-t003:** Results of multilevel moderating effect for testing Hypotheses 4 (model 2).

	OBSE		
	B	SE	β
*Level 2 (N = 82)*			
Residual variances	0.335 **	0.055	
*Level 1 (N = 345)*			
Gender	−0.004	0.006	−0.002
Age	0.024	0.063	0.018
Education	0.002	0.004	0.002
Team tenure	−0.014	0.009	−0.058
Workplace ostracism	−0.501 **	0.089	−0.446 **
Sense of power	0.063	0.075	0.065
Workplace ostracism × Sense of power	0.097 *	0.046	0.093 *
R^2^	0.494		

Note: N_level-1_ = 345, N_level-2_ = 82; Bs represents unstandardized regression coefficients. * *p* < 0.05, ** *p* < 0.01.

## Data Availability

The datasets generated and analyzed during the current study are available from the corresponding author on reasonable request.

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
