# Peer review of "Silence as a Quiet Strategy: Understanding the Consequences of Workplace Ostracism Through the Lens of Sociometer Theory"

_behavsci, 2025, doi:10.3390/bs15081022_

Round 1
Reviewer 1 Report
Comments and Suggestions for Authors
Comments to the Authors:
I have a few general comments followed by some specific suggestions. It would be helpful if the authors could clarify the main research questions of their study. As a reader, I expect to understand why they chose this topic and why they consider it important. We need to know the author’s take on the main reasons for ostracism in the workplace. What is the author’s version?
Author Response
We sincerely appreciate the reviewer’s feedback. However, it appears that the specific comments were not attached. At this stage, we have addressed the general comments to the best of our ability. Should the reviewer provide the detailed comments in the future, we would be glad to incorporate them into our subsequent revision. Please see the attachment for our detailed response.

Reviewer 2 Report
Comments and Suggestions for Authors
Dear Authors,
Thank you for the opportunity to review your manuscript. Please refer to the attached file for detailed comments and suggestions. I hope the feedback will be helpful for your revision.

Author Response
Thank you for your valuable input. We’ve carefully reviewed your suggestions and incorporated them into the revised manuscript. Please find our responses in the attachment.

Reviewer 3 Report
Comments and Suggestions for Authors
thank you for the opportunity to review the manuscript;
here are my comments:
- introduction should be revised to include contextual setting of the research besides mentioning the literature
- authors should improve the recency of their arguments as majority of references are not recent in the introduction.
- authors should embed the theories and not only mentioning them.
theoretical setting should manifest in the hypotheses as well. - justify the methodology
- justify the analytical approach
- provide sample size calculations
- provide CFA or EFA results or measurement model assessment results (reliability and validity measures)
- the authors should better explain the moderation results as the interaction seems to be the merit of their research.
- revise conclusions and discussions after implying the aforementioned issues.
- discussion and conclusions should first discuss each hypothesis and compare with recent literature, and second, conclude and highlight the findings and contributions
- both theoretical and practical implications seem contrived. authors must think critically and provide meaningful implications that can be acted upon rather than abstract ideas.
- ensure the recency of the arguments, is there no newer study than Williams 2001 for example?
authors must maintain a unified tone throughout the research and not switch between academic and conversational tones.
Author Response

(The authors gave the same response as above.)

Reviewer 4 Report
Comments and Suggestions for Authors
The study presents a relevant and innovative contribution by exploring the phenomenon of defensive silence in the workplace through the lens of sociometer theory, proposing a theoretical model that includes the mediation of organization-based self-esteem (OBSE) and the moderation of perceived power. The reconceptualization of silence as a proactive and not merely reactive strategy represents a theoretical advancement with potential to enrich debates in organizational psychology.
1. The theoretical framework is well developed, with a thorough review of both current and classical literature (e.g., Leary et al., 1995; Pierce et al., 1989). The use of sociometer theory is appropriate and well articulated with the constructs under analysis. The hypotheses are logically and coherently derived from the theory presented.
Suggestions for improvement:
-
It would be helpful to more clearly distinguish between emotional mechanisms and the proposed self-regulatory mechanisms throughout the text.
-
Some paragraphs are dense and could be more concise, particularly in the theoretical introduction (lines 39–63).
2. The methodology is robust, with a three-wave study design and data from multiple sources (employees and supervisors). The use of multilevel modeling is justified, given the nested structure of the data. The translation and adaptation of the instruments to the Chinese context were carefully described. The sample is adequate (N = 345), with a good response rate. Measuring variables at different time points helps reduce common method bias.
Points for improvement:
-
Despite the temporal structure, the design is not truly longitudinal. It should be made clearer that this is a time-lagged design, and the limitations of this choice should be discussed.
-
The rationale for using perceived power as a moderator is appropriate, but the operationalization could have been more detailed.
3. The statistical analyses were well conducted (HLM with Mplus 7.4), and the results are clearly presented. The mediation (via OBSE) and moderation (sense of power) were adequately tested, and indirect effects were appropriately interpreted.
Technical suggestions:
-
Include standardized coefficients (β) in the tables or text to facilitate the comparison of effects and enhance interpretation of their magnitude.
-
Mention model goodness-of-fit indicators, if available, as these were not reported and could strengthen the robustness of the proposed model.
-
Clarify a possible inconsistency in the reporting: at one point, the indirect effect is said to be 0.386 and, shortly after, 0.082. Presumably, the former refers to the total effect? This distinction should be clarified to avoid confusion for the reader.
4. The discussion is well articulated, with relevant reflections on the theoretical contributions and practical implications. The inclusion of the supplementary analysis on the (in)effectiveness of silence in reducing ostracism is a strong point, demonstrating a critical and realistic approach.
Recommendations for improvement:
-
Explore more thoroughly the limitations of silence as a coping strategy, especially in collectivist cultures such as China, where preserving interpersonal harmony may influence silence behavior.
-
The practical implications for managers are valid but could benefit from more concrete examples of possible interventions in the organizational context.
While the discussion section addresses the main conclusions of the study and acknowledges some important methodological limitations, it is recommended that the following additional aspects—currently not fully discussed—be considered:
- Weak correlation between OBSE and defensive silence:
Although the effect of OBSE on defensive silence was statistically significant in the multilevel model, the simple correlation between these variables is weak (r = –0.123). It would be useful to discuss this discrepancy, perhaps considering the role of control variables or the nested structure (e.g., team-level variance) in explaining the results.
- Modest magnitude of the moderation effect:
While the interaction between ostracism and sense of power was statistically significant, the interaction coefficient is small (B = 0.097). A brief reflection on the practical relevance of this effect would help contextualize the construct's utility in applied organizational settings.
- Lack of standardized coefficients and variance explained:
Including standardized coefficients and/or measures of explained variance (such as R² or pseudo-R²) would enhance the transparency and interpretability of the results. If such indicators were not computed, it would be helpful to acknowledge this as a limitation.
The limitations section is well delineated. It is worth highlighting the positive reference to the need to replicate the study in different cultural contexts, as well as the recognition of the lack of control for emotional mechanisms (e.g., fear), which limits the clear distinction between emotional and sociometric pathways.
Author Response

(The authors gave the same response as above.)

Round 2
Reviewer 3 Report
Comments and Suggestions for Authors
authors have met my concerns. Kudos!